# Genome-Wide Transcriptome Profiling Reveals the Mechanisms Underlying Hepatic Metabolism under Different Raising Systems in Yak

**DOI:** 10.3390/ani14050695

**Published:** 2024-02-23

**Authors:** Mengfan Zhang, Xita Zha, Xiaoming Ma, Yongfu La, Xian Guo, Min Chu, Pengjia Bao, Ping Yan, Xiaoyun Wu, Chunnian Liang

**Affiliations:** 1Key Laboratory of Yak Breeding Engineering of Gansu Province, Lanzhou Institute of Husbandry and Pharmaceutical Sciences, Chinese Academy of Agricultural Sciences, Lanzhou 730050, China; zmf13664139695@163.com (M.Z.); maxiaoming@caas.cn (X.M.); layongfu@caas.cn (Y.L.); guoxian@caas.cn (X.G.); chumin@caas.cn (M.C.); baopengjia@caas.cn (P.B.); pingyanlz@163.com (P.Y.); 2Key Laboratory of Animal Genetics and Breeding on Tibetan Plateau, Ministry of Agriculture and Rural Affairs, Lanzhou 730050, China; 3Qinghai Province Qilian County Animal Husbandry and Veterinary Workstation, Qilian 810400, China; zhaxita@163.com

**Keywords:** yak meat, liver, lipid metabolism, RNA-seq, fattening

## Abstract

**Simple Summary:**

Yaks are a major economic source for people in the Tibetan Plateau region. Yaks are rich in nutrients, but their low fat content is not conducive to the large-scale promotion of yak meat, so the study of the mechanism of yak lipid deposition is beneficial to the marketing of yak meat. In this paper, the results of transcriptome sequencing analysis of yak liver showed that it could be determined that the expression levels of genes associated with partial lipid deposition were significantly up-regulated during yak fattening. In addition, this study found that the tenderness of yak meat improved during this process. Fattening significantly affects fat deposition in yaks, which may be realized through its effects on lipid metabolic pathways. Therefore, studying the mechanism of lipid deposition in yaks and fattening yaks will improve the quality of yak meat.

**Abstract:**

Yak meat is nutritionally superior to beef cattle but has a low fat content and is slow-growing. The liver plays a crucial role in lipid metabolism, and in order to determine whether different feeding modes affect lipid metabolism in yaks and how it is regulated, we employed RNA sequencing (RNA-seq) technology to analyze the genome-wide differential gene expression in the liver of yaks maintained under different raising systems. A total of 1663 differentially expressed genes (DEGs) were identified (|log2FC| ≥ 0 and *p*-value ≤ 0.05), including 698 down-regulated and 965 up-regulated genes. According to gene ontology (GO) and KEGG enrichment analyses, these DEGs were significantly enriched in 13 GO terms and 26 pathways (*p* < 0.05). Some DEGs were enriched in fatty acid degradation, PPAR, PI3K-Akt, and ECM receptor pathways, which are associated with lipid metabolism. A total of 16 genes are well known to be related to lipid metabolism (e.g., *APOA1*, *FABP1*, *EHHADH*, *FADS2*, *SLC27A5*, *ACADM*, *CPT1B*, *ACOX2*, *HMGCS2*, *PLIN5*, *ACAA1*, *IGF1*, *FGFR4*, *ALDH9A1*, *ECHS1*, *LAMA2*). A total of 11 of the above genes were significantly enriched in the PPAR signaling pathway. The reliability of the transcriptomic data was verified using qRT-PCR. Our findings provide new insights into the mechanisms regulating yak meat quality. It shows that fattening improves the expression of genes that regulate lipid deposition in yaks and enhances meat quality. This finding will contribute to a better understanding of the various factors that determine yak meat quality and help develop strategies to improve yield and quality.

## 1. Introduction

Yaks are a unique domestic animal on the Qinghai-Tibet Plateau [1,2]. They provide the locals with essential living materials as well as an important source of economic activity [3,4]. Yak meat is delicious, high in protein, low in fat, and rich in a variety of vitamins and minerals [5]. However, the meat quality of yak is affected by factors such as rearing methods and living conditions, which result in poor tenderness and a rough taste. This poses a challenge to the overall quality of yak meat and hinders the development of the yak meat industry in the Tibetan Plateau region. The yak is fed primarily by natural grazing, but the climate in the plateau region is harsh and cold, and they experience a cold season for more than six months in a year [6]. During that time, the grassland is low in nutrients, and seasonal changes have altered its nutrient balance [7]. With the change of seasons, the body condition of yaks also displays a vicious cycle of “strong in summer, fat in autumn, thin in winter, and weak in spring”. As a result, the growth and development rate of the yak are also affected. Yaks can be compensated for growth in the warm season when fattening can increase lipid deposition in yaks, improve production performance and meat quality, and restore muscle and other tissues and organs to the best possible condition [8]. Feeding also improves the quality of cattle meat by increasing fat content and tenderness while significantly reducing moisture content. These benefits have been observed in previous studies. Supplemental feeding can effectively increase the content of intramuscular fat [9,10], adjust the tenderness of meat, and provide a new way to improve the quality of yak meat [11,12,13].

The quantity of intramuscular fat (IMF) and the properties of muscle fiber are major determinants of meat quality. IMF content is positively correlated with sensory quality traits such as tenderness, juiciness, and flavor. Several tissues and organs in livestock are closely associated with lipid metabolism, such as the liver, small intestine, and adipose tissue [6]. Along with skeletal muscle and adipose tissue, the liver plays an important role in mammalian lipid metabolism and other processes [14]. It is the central organ of absorption, oxidation, and metabolic transformation of nonesterified fatty acids. In addition, it has enzyme activity for de novo adipogenesis, cytoplasmic storage of triacylglycerol, and the synthesis of fatty acids from glucose and other nonlipid precursors. Lipid metabolism affects not only fat synthesis but also immune function [15]. The liver also participates in the endocrine growth pathway and supplies energy for biological processes, thereby impacting the growth and development of the body [16]. Transcriptome sequencing has emerged as a powerful tool for identifying functional genes involved in lipid metabolism in livestock and poultry livers. Numerous research studies have been conducted in this area, contributing to a better understanding of the molecular mechanisms underlying lipid metabolism in these animals. Asep Gunawan et al. [17] identified a possible candidate gene that controls the composition and metabolism of the sheep liver through transcriptomics analysis of unsaturated fatty acids (FA). Huang et al. [18] compared the liver transcriptome of Jinhua pigs and landrace pigs and found that some differentially expressed genes were involved in the processes of redox, lipid synthesis, and metabolism. Liu et al. [19] conducted a study to investigate the genetic regulatory networks responsible for liver lipid deposition and Aflatoxin B1 (*AGB1*)-induced hepatocyte apoptosis in chickens. They found that lipid deposition in chicken liver is primarily regulated through the PPAR signaling pathway, fatty acid degradation, and fatty acid metabolism, which are involved in lipid metabolism. 

In the present study, we used RNA-seq analysis to identify genes and functional pathways related to lipid metabolism in the liver of yaks under different raising systems. These results could provide valuable information on the mechanisms behind liver lipid metabolism and improved beef quality in yak.

## 2. Materials and Methods

### 2.1. Ethical Statement

All experiments involving animals were authorized by the Lanzhou Institute of Husbandry and Pharmaceutical Sciences of the Chinese Academy of Agricultural Sciences (CAAS); the permit number is No. SYXK-2014-0002.

### 2.2. Animals and Tissue Collection

Twenty healthy male yaks aged four with similar body weight (229.75 ± 28.79 kg) were selected from Datong County, Qinghai Province, and randomly divided into two groups of 10 yaks apiece. The grazing group (group C) was fed free-choice grazing without supplemental feed. The fattening group (group T) was fed a total mixed ration (TMR) that was compounded according to the total energy required for a daily gain of 400 g for 200 kg of beef cattle (composition of TMR; see Appendix A). The study lasted 160 days and began in May, with the first 10 days being predation. Each yak was dewormed before the test and weighed every 30 days before feeding in the fattening group and grazing in the grazing group (see Appendix A). In both groups, the experimental animals underwent 24 h of fasting and an 8-h water deprivation phase following the conclusion of the test period. Three healthy (no apparent physiological abnormalities and no signs of disease) yaks were slaughtered from each of the grazing groups and fattening groups, and their livers were isolated and extracted as test samples (total of 6) for transcriptome sequencing. The liver tissue was sectioned into small pieces and rapidly frozen in liquid nitrogen.

### 2.3. RNA Extraction, Library Preparation, and Sequencing

Total RNA was extracted using a Trizol reagent (Transgen Biotech, Beijing, China). Thermo Scientific NanoDrop 2000c (ThermoFisher Scientific Inc., Waltham, MA, USA) was used to determine the concentration and purity of the extracted RNA. The integrity of the RNA was detected by 1% agarose gel electrophoresis. The total amount of RNA should be ≥1 μg, the concentration ≥ 50 ng/μL, and the value of OD 260/OD 280 should be between 1.8 and 2.2 [20]. An oligomer magnetic bead (dT) approach was used to enrich Poly A mRNA from total RNA (6 samples) [21,22]. A cDNA library was constructed by synthesizing cDNA from random hexamers, purifying cDNA, and amplifying it by PCR. A NEBNext^®^ Ultra RNA Library Prep Kit for Illumina was used to perform RNA-seq library preparation and, after library inspection, qualified. Finally, paired-end sequencing of different libraries was accomplished using Illumina sequencing.

### 2.4. Data Quality Control and Reference Genome Comparison

Illumina high-throughput sequencing results were preprocessed to produce clean reads and ensure data reliability. To determine which genes were transcribed by the sequenced fragments, HISAT2 was used to map clean reads to the reference genome (BosGru_v2.0) [23]. The gene expression analysis was calculated by counting the unique matches between reads.

### 2.5. Screening of DEGs

A normalization of the FPKM (expected number of fragments per kilobase of transcript sequence per million base pairs sequenced) into matched reads was performed by Cufflinks. With feature counts in subread, genomic expression levels are analyzed individually for each sample. Different samples were analyzed using DESeq2 (v 1.16.1), and DEGs genes were identified as significant genes with |log2FC| ≥ 0 and *p*-value ≤ 0.05.

### 2.6. GO and KEGG Enrichment Analysis

ClusterProfiler software (v 3.4.4) was used to determine the functional categories and biological functions of differential genes based on the Gene Ontology (GO) functional enrichment analysis and the Kyoto Encyclopedia of Genes and Genomes (KEGG) pathway enrichment analysis. The enrichment analysis was based on the hypergeometric distribution principle. DEGs were enriched for the GO and KEGG pathways, with a threshold of *p* < 0.05 for significant enrichment.

### 2.7. Validation of Candidate Gene Results by PT-qPCR

Eight genes were randomly selected for RT-qPCR to verify the accuracy of the transcriptome sequencing data. RT-qPCR was conducted on the same RNA samples that were used for RNA-seq. The experimental primer pairs are listed in Appendix A. The reaction system was 20 µL. The procedure involved 45 cycles of pre-denaturation at 95 °C for 3 min; denaturation at 95 °C for 10 s, annealing at 60 °C for 10 s, and 72 °C for 10 s. There were triplicates in all experiments. The 2^−∆∆CT^ method was used to analyze the changes in relative gene expression.

## 3. Results

### 3.1. Analysis of Transcriptome Sequencing Quality

A total of 27.97 million original reads were obtained. After removing connectors, N-containing reads, and low-quality reads, the clean reads obtained in each group exceeded 6.46 G. The qualifying rates were 94.96%, 95.06%, 95.21%, 93.53%, 94.72%, and 95.09%, respectively, all above 90%. The Q30 values ranged from 93.53% to 95.21%, meeting the requirement of more than 90% for the Q30 sequence, and the GC content of six samples ranged from 49.27% to 50.57%. The reads were of good quality and could be further analyzed. The basic statistics of liver RNA-seq reads in the fattening and grazing groups are shown in Appendix B (see Appendix B Table A1).

Using HISAT2 software (v 2.0.5), clean reads from the sample were quickly and accurately mapped to the reference yak genome (BosGru_v2.0), and the mapping ratio ranged from 88.99% to 94.54%. The results of the comparisons are shown in Table 1. Reference genomes are counted in different regions (exons, introns, and intergenic regions). Generally, the longer the chromosome, the more reads can be distributed to that chromosome. According to the comparison of the six samples, the exon region had the highest percentage of reads (above 83%), while a small number of reads were in the intron region and the intergenic region (about 6% and 8%, respectively); the distribution of data is normal, which indicates good sequencing data quality. A few read-matching introns may be derived from retained introns. In contrast, those matching intergenic groups may result from contamination with DNA or ncRNA fragments, and an inadequate annotation of genes may also cause the problem.

### 3.2. Analysis of the Level of Gene Expression

The expression levels of the sample genes were depicted using box-and-line plots (see Appendix A), which show that the dispersion is low, the reproducibility is good, and the overall expression is good. From the FPKM density distribution graph (see Appendix A), it can be seen that the density distribution curves of the samples in each group are more consistent, and the combination of the two graphs can indicate that the samples have good expression levels. When comparing the FPKM values of six samples, overall gene expression levels were similar (see Appendix B Table A2).

As shown in the correlation heatmap (see Appendix A), clustering the different replicate samples in each group, in each group of replicate samples, the Pearson correlation coefficient squared (R^2^) is greater than 0.88, close to 1. This indicates a high level of biological replication and a high level of similarity in the expression pattern between the samples. The principal component analysis (PCA) plot shows (see Appendix A) that the samples were scattered between groups and clustered within groups, indicating that the samples in this experiment were reasonably selected and the samples were reproducible.

### 3.3. Statistics and Cluster Analysis of Differentially Expressed Genes

DESeq2 software (v 1.16.1) was used to analyze differential transcription between the two combinations based on the good repeatability of biological samples. According to the differential screening conditions (|log2FC| ≥ 0 and *p*-value ≤ 0.05), 1663 differentially expressed genes were identified after comparison with the database. Among them, 965 differentially expressed genes were up-regulated, while 698 differentially expressed genes were down-regulated in the control group. On the volcano map (Figure 1a), the overall distribution of differential genes as well as the distribution of differential genes in each comparison pair are shown. All differential genes from the test groups were collected as differential gene sets; FPKM values were used in the cluster analysis to better understand the gene expression patterns of the grazing group and the fattening group. The results of the cluster analysis in the figure show that the genes differentially expressed in the grazing and fattening groups were grouped into one class each, those genes with up-regulated expression were grouped into one class, and those with down-regulated expression were grouped into one class (Figure 1b).

### 3.4. GO Functional Enrichment Analysis

This study used ClusterProfiler software (v 3.4.4) to annotate the functions of significantly differentially expressed genes, with *p* less than 0.05 as the threshold for significant enrichment. The selected DEGs were enriched to 768 GO terms and significantly enriched to 13 GO terms. It contains 10 molecular function classes, 2 cellular component classes, and 1 biological process class (see Appendix A). The top 10 terms with the highest significance were plotted from each functional classification as a graph (Figure 2a). The up-regulated genes are significantly enriched in the oxidation-reduction process, the extracellular region, calcium ion binding, oxidoreductase activity acting on paired donors, with the incorporation or reduction of molecular oxygen, etc. (Figure 2b). Down-regulated genes are significantly enriched in the oxidation reduction process, cofactor binding, oxidoreductase activity, iron ion binding, etc. (Figure 2c).

### 3.5. KEGG Pathway Enrichment Analysis

The analysis of the KEGG pathway shows that DEGs were significantly enriched in 26 pathways, including chemical carcinogenesis, retinol metabolism, and ECM-receptor interaction (Figure 3). Among them, fatty acid degradation, the PPAR signaling pathway, the ECM-receptor interaction, and the PI3K-Art signaling pathway are closely related to lipid metabolism (Figure 4 and Figure 5). Sixteen lipid-metabolism-related DEGs were identified in the livers of the fattening group and the grazing group; among these genes, 4 genes were up-regulated and 12 genes were down-regulated (see Appendix A). These 16 genes were mainly enriched in the four pathways mentioned above.

### 3.6. Validation of Candidate Gene Analysis Results by RT-qPCR

In order to further validate the result of RNA-seq, eight genes (*LAMA2*, *IGF1*, *APOA1*, *FABP1*, *SLC27A5*, *CPT1B*, *HMGCS2*, and *PLIN5*) were chosen to detect expression in the liver by RT-qPCR. As shown in Figure 6, *LAMA2*, *IGF1*, *APOA1*, and *FABP1* were up- regulated, and *SLC27A5*, *CPT1B*, *HMGCS2*, and *PLIN5* were down-regulated in the livers of fattening yaks, which is consistent with RNA-seq results.

## 4. Discussion

The liver is the main site of metabolism and plays an important role in fat metabolism. In this study, the comparison of RNA-seq techniques revealed 1663 differentially expressed genes (DEGs) in yaks under two different feeding methods, of which 698 genes were down-regulated and 965 genes were up-regulated. The reliability of the transcriptomic data was verified using qRT-PCR. By GO and KEGG enrichment analysis of DEGs, adipose metabolism-related genes were screened (*APOA1*, *FABP1*, *EHHADH*, *FADS2*, *SLC27A5*, *ACADM*, *CPT1B*, *ACOX2*, *HMGCS2*, *PLIN5*, *ACAA1*, *IGF1*, *FGFR4*, *ALDH9A1*, *ECHS1*, *LAMA2*). The above genes are mainly involved in fatty acid degradation, lipid metabolism, etc. The differences in feeding conditions could make the lipid metabolism of yaks different. Combined with the changes in meat quality, fattening can improve the expression of the genes regulating lipid deposition in yaks and enhance meat quality.

DEGs detected in the livers of yaks between the fattening group and the grazing group in this study were enriched in many pathways related to lipid metabolism, such as ECM-receptor interaction, the PPAR signaling pathway, fatty acid degradation, and the PI3K-Akt signaling pathway, which regulate the yak lipid metabolism process together. Among them, there is a close correlation between the PPAR signaling pathway and fatty acid degradation. The correlation between the PPAR signaling pathway and fatty acid degradation is mainly manifested in the role of PPARs in the regulation of lipid metabolism and adipocyte differentiation. Huang et al. [24] and Huang et al. [18], respectively, confirmed that key genes related to lipid metabolism and lipid generation in the liver tissues of cattle or pigs are enriched in the PPAR signaling pathway. A total of 11 of the 16 genes associated with lipid metabolism screened in this assay were significantly enriched in the PPAR signaling pathway. The PPAR signaling pathway is a key pathway closely related to lipid metabolism, lipid differentiation, and other functions [25]. The peroxisome proliferator-activated receptor α (*PPARα*) is closely related to the Perilipin 5 gene (*PLIN*5) and their interaction in the regulation of lipid metabolism and deposition [26]. In combination with *CGI-58*, the *PLIN5* gene inhibits lipocatabolic processes mediated by adipose triglyceride lipase (*ATGL*) [27]. As per the findings of Wang et al. [27], the liver lipid content would be reduced when the *PLIN5* gene was absent. Fatty acid desaturase 2 (*FADS2*), a member of the fatty acid dehydrogenase family, is a key rate-limiting enzyme during polyunsaturated fatty acid metabolism. It plays an important role in maintaining the correct structure of the membrane and regulating fatty acid metabolism in vivo. Acetyl-CoA acyltransferase 1 *(ACAA1*) is an essential enzyme found downstream of the PPAR pathway. It serves as a crucial component in the synthesis and transport of fatty acids, catalyzes the synthesis of esterified cholesterol from free cholesterol and long-chain fatty acids, and plays an important role in the fatty oxidation process [28]. In mammals, the peroxisome proliferator-activated receptor (*PPAR*) controls the expression of *CPT1* mRNA and protein to some extent [29]. As an important isoenzyme of *CPT1*, the Carnitine Palmitoyltransferase 1B gene (*CPT1B*) plays an important role in the regulation of fatty acid oxidation in vivo and helps to control long-chain fatty acid transport into mitochondria and their metabolism, thus affecting the content of fatty acids [30,31].

Cyl-CoA Dehydrogenase Medium Chain (*ACADM*) catalyzes the β-oxidation of medium-chain fatty acids [32], and the deficiency of this gene can cause fatty acid metabolism disorders and liver function abnormalities [33]. 3-Hydroxy-3-Methylglutaryl-CoA synthase 2 (*HMGCS2*) provides lipid-derived energy to hepatocytes, and up-regulated expression of the *HMGCS2* gene is associated with fatty acid oxidation induced by high-fat diets [34]. Apolipoprotein A1 (*APOA1*) is the major protein component of the plasma high-density lipoprotein involved in the reverse transport of cholesterol to the liver through cholesterol acyltransferase. Liu et al. [35] found that the *APOA1* gene is related to the growth and development of porcine fat and may be a candidate gene for regulating lipid deposition. Therefore, it is speculated that this gene affects lipid deposition by regulating the liver. The Fatty Acid Binding Protein 1 (*FABP1*) gene has a high binding capacity with long-chain fatty acids and mainly regulates various lipid signals to encode and participate in lipid-mediated signaling pathways and metabolic homeostasis [36]. The *FABP1* gene is a regulator of TAG (triacylglycerol) and VLDL (very low density lipoprotein) in the liver [37]. Xiong et al. [38] analyzed the effects of the *FABP1* gene on the biological function of porcine intramuscular adipocytes based on RNA-seq. They found the mechanism of the involvement of the *FABP1* gene in the regulation of lipid deposition and lipid metabolism. The *ACOX2* gene encodes branched-chain acyl-CoA oxidase, which is involved in the degradation of long-chain fatty acids and intermediates of bile acid in the peroxisome and plays a vital role in lipid metabolism [39].

The ECM receptor interaction pathway has direct or indirect effects on cell adhesion, migration, and other activities [40]. This pathway is thought to play an essential role in the regulation of the intramuscular adipocyte differentiation, lipid synthesis, and metabolism of intramuscular adipocytes and influence IMF content [41]. Laminin subunit alpha 2 (*LAMA2*) can affect the PI3K-AKT pathway and is also a downstream effector of the ECM receptor pathway [42]. *LAMA2* gene mutations have been shown to cause congenital muscular dystrophy in dogs and mice [43,44].

The fatty acid degradation pathway is closely related to lipid deposition. The fatty acid degradation pathway can reduce lipid levels and fat accumulation by promoting fat oxidation [45]. The genes enriched in fatty acid degradation (*CPT1B*, *ACADM*, *ECHS1*, *EHHADH*, *ACAA1*, and *ALDH9A1*) screened in this study show a trend of down-regulation of their expression, which suggests that fattening can promote fat deposition. English et al. [46] found that differential fat genes in subcutaneous fat from male calves were significantly enriched in the fatty acid degradation pathway under different dietary conditions. Mitochondrial fatty acid β-oxidation is the main pathway of lipid degradation [47]. Enoyl-CoA Hydratase, Short Chain 1 (*ECHS1*) is a key enzyme in mitochondrial fatty acid β-oxidation [48]. Studies have shown that the Aldehyde dehydrogenase 9 family member A1 (*ALDH9A1*) gene plays an important role in lowering blood lipids and promoting fatty acid metabolism in rats [49]. Genome-wide association analysis in several pig populations found that ALDH9A1 was correlated with the fatty acid content in the muscle and abdominal adipose tissue of pigs [50,51]. Enoyl-CoA Hydratase and 3-Hydroxyacyl CoA Dehydrogenase (*EHHAD*) is part of the fatty acid β-oxidation pathway that can be induced by PPARα activation [52]. Studies in Holstein dairy herds have shown that *EHHAD* is a functional gene that potentially affects the composition and content of fatty acids in milk [53].

The PI3K-Akt signaling pathway promotes lipid biosynthesis and inhibits lipolysis [54,55,56]. The decrease in the level of insulin-like growth factor 1 (*IGF-1*) induces a decrease in PI3K-Akt signaling, which further affects lipid deposition [57,58]. FGFR4 inhibition not only has effects on lipid catabolism and secretion but also down-regulates de novo adipogenesis by reducing the expression of genes involved in triglyceride synthesis (such as *SREBP1c*, *ACC*, *FAS*, and *DAGT1*) [59].

The tenderness of meat is one of the important indexes to evaluate the quality of meat products, and the fat content is the related index of tenderness that affects the quality of meat. A previous study of our research group [60] showed that supplementary feeding improved the shear force of yak meat; it was indicated that muscle tenderness in the fattening group was improved, in turn deducing an increase in intramuscular fat (see Appendix A). The DEGs obtained in this study are closely involved in lipid metabolism (*APOA1*, *FABP1*, *IGF1*, *LAMA2*, *EHHADH*, *FADS2*, *SLC27A5*, *ACADM*, *CPT1B*, *ACOX2*, *HMGCD2*, *PLIN5*, *ACAA1*, *FGFR4*, *ALDH9A1*, *ECHS1*), which are significantly enriched in the signaling pathway engaged in lipid metabolism. These genes were significantly expressed in the liver of fattened yaks (*p* < 0.05). At the same time, the expression level of these genes also changed compared to the grazing group; these genes enriched the fat of the yak and contributed to the improvement of meat tenderness in the yak, which is consistent with the previous research results of our group and demonstrated the influence of fattening on the meat tenderness of the yaks from the gene level. The differentially expressed genes selected by transcriptome analysis in this study can be associated with lipid deposition and used as candidate genes to improve yak meat quality and as biomarkers for yak breeding.

## 5. Conclusions

In this study, we used RNA-seq technology to compare the liver tissues of yaks from the grazing group and the fattening group and screened several differentially expressed genes related to lipid metabolism, for example, *FGFR4*, *ALDH9A1*, *ECHS1*, etc. Through GO term enrichment and KEGG pathway enrichment analysis, differentially expressed genes were enriched in several pathways closely related to lipid metabolism. However, this study only explained the mechanism of hepatic lipid metabolism at the transcriptome level. Further joint analysis at the metabolic level is needed in the future. It provides a theoretical basis for exploring the mechanism of yak lipid metabolism and improving the quality of yak meat.

## Figures and Tables

**Figure 1 animals-14-00695-f001:**
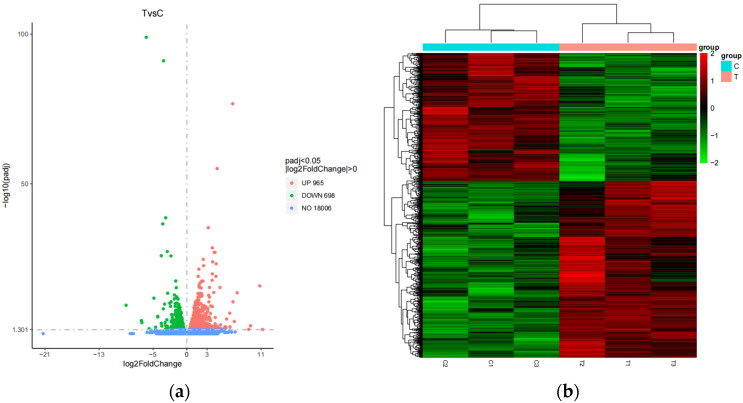
Statistics of differentially expressed genes in the grazing group and the fattening group: (**a**) Volcano map of the number of differentially expressed genes. (**b**) Cluster analysis of the number of differentially expressed genes.

**Figure 2 animals-14-00695-f002:**
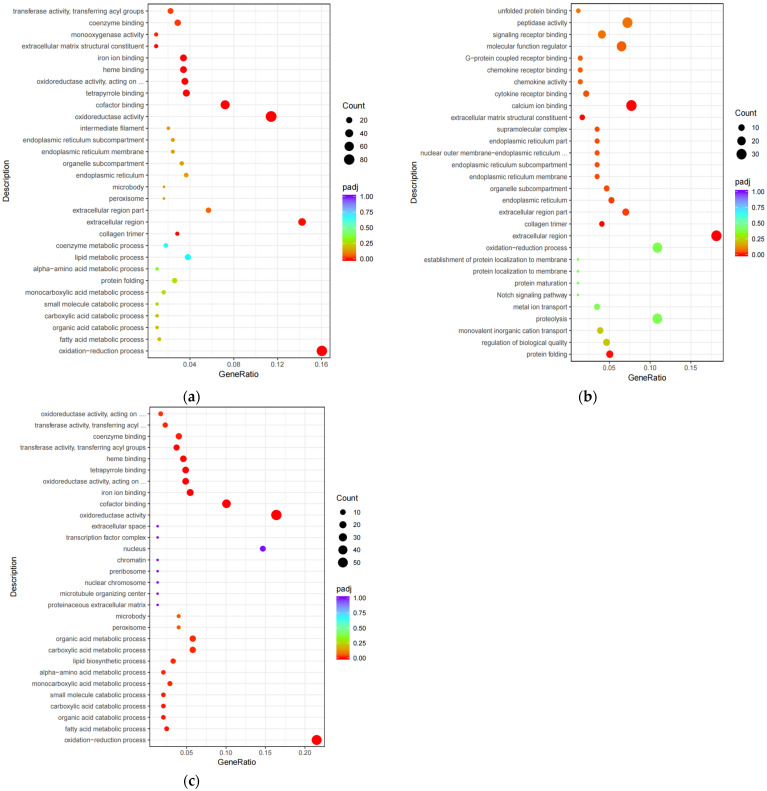
GO item enrichment analysis: (**a**) Scatterplot diagram of differentially expressed gene GO enrichment analysis; (**b**) scatterplot of GO enrichment analysis of up-regulated differentially expressed genes; (**c**) scatterplot of GO enrichment analysis of down-regulated differentially expressed genes.

**Figure 3 animals-14-00695-f003:**
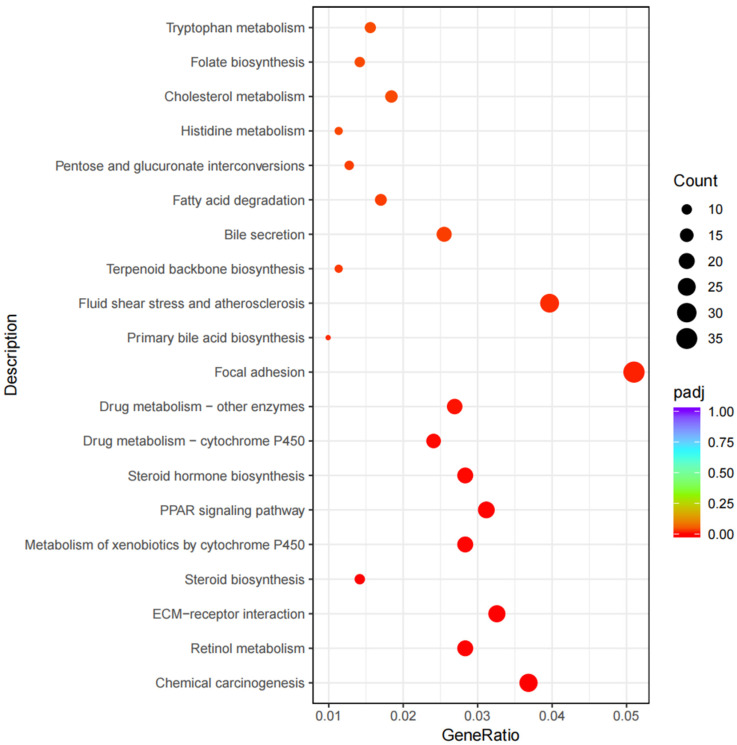
Scatterplot diagram of KEGG enrichment analysis of differentially expressed genes.

**Figure 4 animals-14-00695-f004:**
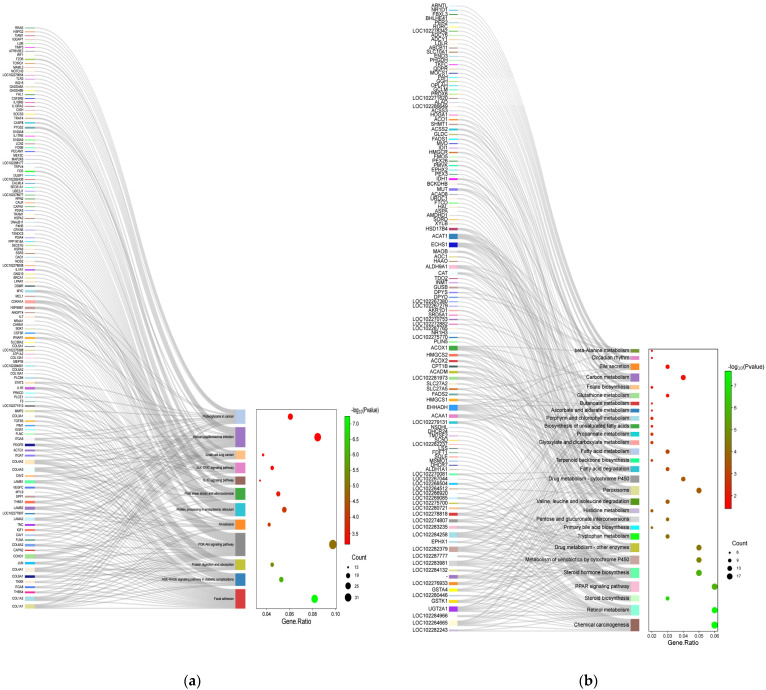
Sankey map of differentially expressed genes: (**a**) Sankey map of the KEGG enrichment analysis of up-regulated differentially expressed genes; (**b**) Sankey map of the KEGG enrichment analysis of down-regulated differentially expressed genes.

**Figure 5 animals-14-00695-f005:**
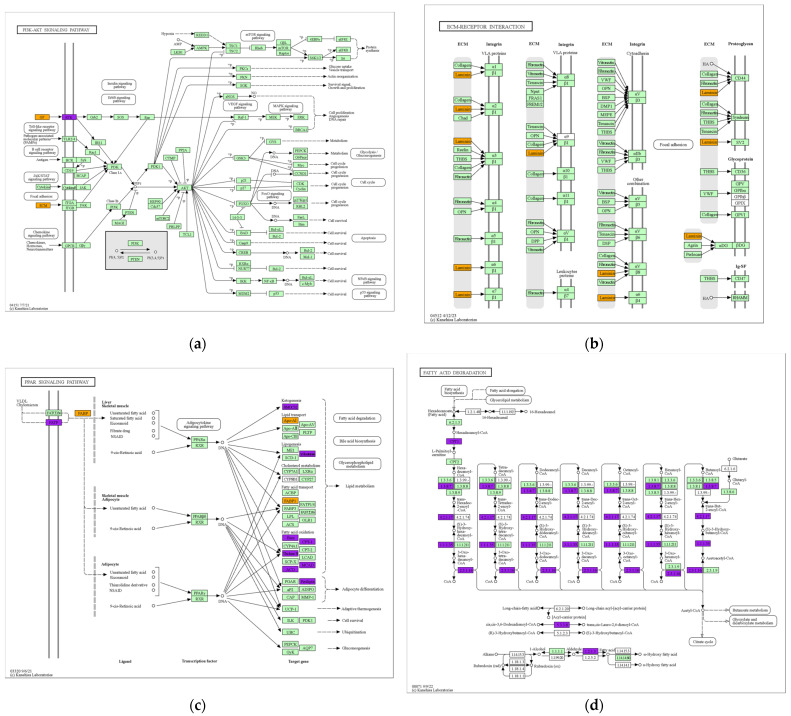
Significant enrichment pathway of differentially expressed genes: (**a**) PI3K-Akt signaling pathway; (**b**) ECM-receptor signaling pathway; (**c**) PPAR signaling pathway; (**d**) fatty acid degradation pathway.

**Figure 6 animals-14-00695-f006:**
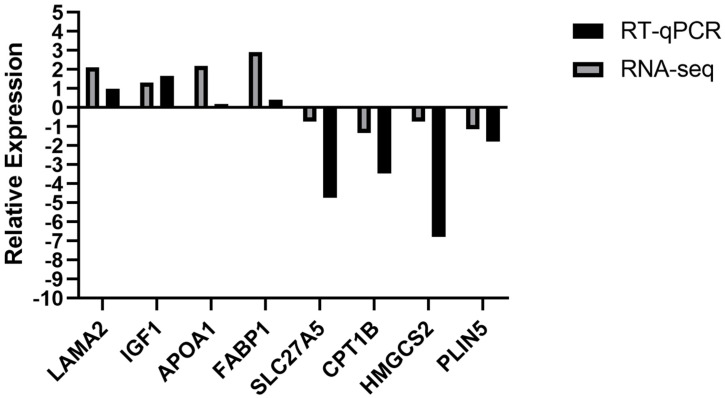
Validation of RNA-Seq results via RT-qPCR.

**Table 1 animals-14-00695-t001:** The results compared to the reference genome.

Sample	Total Reads	Total Mapped	Unique Mapped	Multi Mapped	Positive_Map	Negative_Map
C1	46,643,568	41,508,351 (88.99%)	40,274,319 (86.34%)	1,234,032 (2.65%)	20,142,574 (43.18%)	20,131,745 (43.16%)
C2	45,183,264	42,522,445 (94.11%)	41,259,349 (91.32%)	1,263,096 (2.8%)	20,628,309 (45.65%)	20,631,040 (45.66%)
C3	44,915,026	42,314,469 (94.21%)	41,022,378 (91.33%)	1,292,091 (2.88%)	20,506,370 (45.66%)	20,516,008 (45.68%)
T1	46,353,528	43,821,417 (94.54%)	42,505,095 (91.7%)	1,316,322 (2.84%)	21,234,481 (45.81%)	21,270,614 (45.89%)
T2	43,810,052	41,220,856 (94.09%)	39,888,411 (91.05%)	1,332,445 (3.04%)	19,909,588 (45.45%)	19,978,823 (45.6%)
T3	43,082,038	40,613,909 (94.27%)	39,399,963 (91.45%)	1,213,946 (2.82%)	19,691,779 (45.71%)	19,708,184 (45.75%)

## Data Availability

The data presented in this study are openly available in Sequence Read Archive at https://www.ncbi.nlm.nih.gov/sra (accessed on 10 December 2023), reference number PRJNA1031991.

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
