# Peer review of "Genome-Wide Transcriptome Profiling Reveals the Mechanisms Underlying Hepatic Metabolism under Different Raising Systems in Yak"

_animals, 2024, doi:10.3390/ani14050695_

Round 1

Reviewer 1 Report (Previous Reviewer 1)

Comments and Suggestions for Authors

The authors have answered all my questions and did a proper revised.

Author Response

Dear Reviewer

Thank you for your positive and encouraging comments on our manuscript. Once again, I would like to thank you for all the comments, your careful review and pointing out the shortcomings in the manuscript have opened up the minds of our authors to a more refined way of thinking. It is your comments that have further improved my manuscript. Thank you for your comments and suggestions.

Sincerely

Mengfan Zhang

Reviewer 2 Report (Previous Reviewer 3)

Comments and Suggestions for Authors

Comment: This manuscript had been well organized, and I approve of its publication. But the discussion should be improved, in addition, the first paragraph should rewrite and including the key findings. Revise the conclusion section and mention the limitation and further future ideas regarding the hypothesis.

Author Response

Dear Reviewer

Thank you for your acknowledgment of this study. We feel great thanks for your professional review work on our paper.  We have carefully read your comments and have revised the discussion and conclusions of the manuscript based on your comments. This can be seen in more detail in the revised manuscript. Thank you again for your careful review, it is your suggestions that have improved the quality of my paper.

Sincerely

Mengfan Zhang

Reviewer 3 Report (New Reviewer)

Comments and Suggestions for Authors

The paper deals with the transcriptome profiling in liver in yak. The article is well made and brings new knowledges. Some minor mostly formal errors must be corrected.

 Throughout the manuscript parts of the text are in red, why.

Delete rr. 27-34 (The yak….acid metabolism). It only repeats generall knowledges, not suitable in Abstract.

R. 68, twice point.

Rr. 75-6, delete The sensory…content, it repeats the previous sentence.

R. 143, explain FPKM.

R. 175, probably intergenic, not intergenomic.

R. 221 software˽to; r. 275 receptor˽alfa; r. 305 FABP1 italic.

Delete rr. 263-6 (In order…was revealed.)

Add legend to Tables A1, A2. They should be better described.

The main question addressed by the research is to describe the transcriptom in liver connected with lipid metabolism in yak.

The topic is original and relevant in the field. The species chosen is specific for China and other countries in Himalayas. The papers dealing with metabolism in yak are rare, it goes more for transcriptomics.

The research adds relevant information on yak metabolism etc., they are very scarce, compared with other published material.

The authors may be should inform that the animals were slaughtered, I do not assume the biopsy.

The conclusions are consistent with the evidence and arguments presented and they address the main question posed.

The count of references is rather high, but I would not change it. 

Author Response

Dear Reviewer

We thank you for the positive and constructive comments regarding our paper, and we sincerely appreciate your feedback, which would help to improve the quality of our manuscript.

  1. Throughout the manuscript parts of the text are in red, why.

We would like to express our gratitude to the reviewers for bringing this matter to our attention. Because this manuscript was resubmitted with revisions, the red lettering is a trace of the revisions retained, and we thank you very much for raising this issue.

  1. Delete rr. 27-34 (The yak….acid metabolism). It only repeats generall knowledges, not suitable in Abstract.

We sincerely appreciate the valuable comment. We have deleted this part according to your suggestion. Some modifications have been made to fit the context. The specific modifications are as follows: Yak meat is nutritionally superior to beef cattle but has a low fat content and is slow-growing. The liver plays a crucial role in lipid metabolism, and in order to determine whether different feeding modes affect lipid metabolism in yaks and how it is regulated. The aim of this study was to investigate the effects of feeding modes on lipid metabolic processes and their regulatory mechanisms in yaks.

  1. 68, twice point.

We were really sorry for our careless mistakes. Thank you for your reminder.

  1. 75-6, delete The sensory…content, it repeats the previous sentence.

Thanks for your careful checks. We are sorry for our carelessness. Based on your comments, we deleted duplicate sentences.

  1. 143, explain FPKM.

We think this is an excellent suggestion. We have added an explanation of FPKM. FPKM means expected number of Fragments Per Kilobase of transcript sequence per Millions base pairs sequenced

  1. 175, probably intergenic, not intergenomic.

We feel sorry for our carelessness. In our resubmitted manuscript, the typo is revised. Thanks for your correction.

  1. 221 software˽to; r. 275 receptor˽alfa; r. 305 FABP1 italic.

Thanks for your careful checks. We are sorry for our carelessness. Based on your comments, we have made the corrections in the manuscripts.

  1. Delete rr. 263-6 (In order…was revealed.)

We think this is an excellent suggestion. We have delete rr. 263-6. Some modifications have been made to fit the context. Details can be found in the Discussion of the revised draft

  1. Add legend to Tables A1, A2. They should be better described.

We sincerely thank you for careful reading, we think this is an excellent suggestion. Based on the reviewers' comments. we have added legends for Tables A1, and A2. The specific modifications are as follows.

(Sample: sample name; Raw_reads: number of reads in the raw data; Clean_reads: number of reads in the raw data after filtering; Clean_bases: number of bases in raw data after filtering; Error_rate: overall sequencing error rate of the data; Q20: Percentage of bases with Phred value greater than 20 out of total bases; Q30: Percentage of total bases with Phred value greater than 30; GC pct: percentage of G and C in clean reads for all four bases.

C1: The first yak in the grazing group; C2: The second yak of the grazing group; C3: The third yak in the grazing group; T1: The first yak in the fattening group; T2: The second yak of the fattening group; T3: The first yak of the fattening group.)

  1. The authors may be should inform that the animals were slaughtered, I do not assume the biopsy.

Thank you very much for pointing out our shortcomings. We have added it to the Animals and tissue collection. The details are as follows: (Three healthy (no apparent physiological abnormalities and no signs of disease) yaks were slaughtered from each of the grazing groups and fattening groups and their livers were isolated and extracted as test samples (total of 6) for transcriptome sequencing.)

Sincerely

Mengfan Zhang

Reviewer 4 Report (New Reviewer)

Comments and Suggestions for Authors

The paper presents interesting content concerning yaks, and more specifically  is an attempt to clarify some genetic mechanisms for regulating meat quality. The authors have provided sufficient background for the research and also have very well defined their aim. Generally, the results are well presented with rich graphic material.  However, in my opinion, the there is a serious shortcoming in the experimental design and this is the very small sample that is used for the genetic analysis. The initial number of animals per group is low for genetic analysis ( n=10) and further the authors have described that they have used 3 hepatic samples per group. I am quite aware, that there are papers presenting results from genetic analysis with low sample size, however 3 is extremely low for any sound conclusions. 

Author Response

Dear Reviewer

Thank you for your positive comments and valuable suggestions on our article. Regarding the sample size issue, you mentioned. We will answer it in two ways. Firstly, yaks are still mainly naturally grazed, thus yak sampling costs are high. Secondly, the sample size of three yaks satisfies the basic requirement of biological replication to some extent. In addition, significant clustering between the two sets of samples was found through principal component analysis (PCA). This means that there are obvious differences between the two groups of samples at the transcriptome level. Consequently, three parallel samples (without increasing the number of parallel samples) can also basically meet the requirements of this experiment, and more obvious results can be obtained to reveal the differences between the two groups. However, we also believe that the small sample size is a shortcoming of our study, and we hope that we can increase the sample size as much as possible in future studies. Thank you again for your careful review and valuable comments!

Sincerely

Mengfan Zhang

This manuscript is a resubmission of an earlier submission. The following is a list of the peer review reports and author responses from that submission.

Round 1

Reviewer 1 Report

Comments and Suggestions for Authors

The study employs RNA-seq technology to investigate the liver transcriptome of yaks under different feeding patterns, shedding light on how these patterns influence yak lipid metabolism. Through the analysis of differentially expressed genes (DEGs), the research identifies key genes related to lipid metabolism in yaks. This research contributes valuable knowledge for understanding fat metabolism in the liver of yaks. 

Major comments:

1. (line 107) I recommend providing the ingredients and nutritional composition of the grass and supplemental feed. This could give us a clear difference in the nutritional value between the grazing and feed.

2. (line 109) It is mentioned that the yaks have been weighed every 30 days, but there is no growth performance data in the results.

3. (line 112) What kind of samples you cut? Please make it specific.

4. (line 114) What is the definition of healthy yaks? Usually, the samples should be taken from animals with similar body weights to reduce bias.

5. (line 324) Did you use RT-qPCR or Western blot to verify the genes listed here (EHHADH, FADS2, SLC27A5, etc.)? I recommend using RT-qPCR and Western blot to confirm the gene and protein expression. Sometimes, the results of RNA-seq are not by the RT-qPCR, and the protein is the actual functional element that impacts the cell metabolism.

6. (line 342) The tested samples are liver samples, and it cannot be concluded that the intramuscular fat content of the yaks increased. It is a hypothesis that was not supported by your data. I recommend testing the intramuscular fat content of longissimus dorsi or other muscle samples from these two groups.

 7. Figure 2 in the manuscript is not clear. The number of padj cannot be seen.

Comments on the Quality of English Language

1. (line 67) the properties muscle fiber should be 'the properties of muscle fiber'

2. (line 75) synthsis should be 'synthesis'

3. (line 81) which mostly used should be ' which is used'

4. (line 83) sequencing study lipid metabolism should be 'sequence to study'

5. (line 85) there are two 'of'

6. (line 155) could be further analysis -- analyzed

7. (line 162) the more Reads ' can be' distributed to that chromosome

8. (line 244) deteced -- 'detected'

9. (line 250) inliver -- 'in the liver'

10. (line 278) for regulate fat deposition -- 'regulating'

11. (line 284) analysed -- 'analyzed'

Reviewer 2 Report

Comments and Suggestions for Authors

1.     There are many grammatical and formatting errors in this manuscript. It is difficult for read and verbose. Such as the sentences in Line 22-23, Line 27-28, Line 49-50, Line 67, Line 82-84, Line 112, etc.

2.     This sentence in Line 27-28 is incomplete.

3.     Line 39: The previous section did not mention any parameters related to tenderness, and it is inappropriate to mention in the summary that fattening can improve muscle tenderness.

4.     The word deposited in Line 40 is incorrect.

5.     Add “of” after properties in Line 67.

6.     Delete “of” in Line 85.

7.     Delete “and fatty acid degradation” in Line 92.

8.     Since it is an experiment evaluating and comparing different feeding modes, why not provide dietary information, such as its composition and nutritional level in Materials and Methods part?

9.     What’s the meaning of sentence in Line 107-108?

10.  ……

11.  Please check the manuscript carefully to revise every grammatical/editorial error. It is suggested to provide an English editing certificate.

12.  Follow more accurately the Guide for ANIMALS.

Comments on the Quality of English Language

 Please check the manuscript carefully to revise every grammatical/editorial error. It is suggested to provide an English editing certificate.

Reviewer 3 Report

Comments and Suggestions for Authors

Dear Authors, in the paper “Genome-wide transcriptome profiling reveals the mechanisms underlying fat deposition under different raising systems in yak”, which is very interesting and the results are important for yak selection. moreover, the liver as an important organ for fat deposition, the authors tried to annotate the DEGs both for yak fat deposition and meat quality, which is interesting. However, I have a few comments or advice which can be considered.

Why the author said tenderness of the meat was improved when yaks were fattened, were there some literatures could support it.

In the abstract, why the authors get a conclusion that fattening can improve the tenderness of yak meat, just because the 16 DEGs enrichment in lipid metabolism? So, I don’t think this is result can be acceptable.

Some sentences should be modified like “Qinghai-Tibet Plateau and its surrounding high-altitude areas, where the living environment is poor”. Which means all these areas?

In the materials, 20 healthy male yaks were performed, however no age information, and why the author just chose 3 healthy yaks in each group?

Lack of RNA RIN value of the six samples.

In the method, where is the reference genome, did you use bosTau9? Therefore, in the results, the figure 4, so many genes are not annotated which with LOC prefix, if these genes without any annotation, how did the author exactly know their GO function, therefore, i suggest to do blast first for these LOC genes.

Some sentences in the discussion were poorly organized, so some of this part should be rewritten. Like Line 323-326 “The DEGs obtained in this study are closely involved in lipid metabolism, among which EHHADH, FADS2, SLC27A5, ACADM, CPT1B, ACOX2, HMGCD2, PLIN5, ACAA1, FGFR4, ALDH9A1, ECHS1 and other genes were significantly enriched in the signaling pathway involved in lipid metabolism, and these genes were significantly expressed in the liver of fattened yaks. I suggest to change the word “other genes”, because there are so many genes related to fatty synthesis and metabolism, and also for fat deposition, which is totally different progress.

Reviewer 4 Report

Comments and Suggestions for Authors

In the study "Genome-wide transcriptome profiling reveals the mechanisms underlying fat deposition under different raising systems in yak" it was presented the liver transcriptome of yaks after traditional feeding as grazing and the second experimental group after fattening. These two transcriptomes were compared and the differentially expressed genes were pinpointed and this observation the authors would like to transfer to the meat yak tenderness. However, in the manuscript are multiple errors:

1. the feeding in the methods is very poor described see lines 107-108 what does mean: containing 200 kg beef cattle with 400 daily weight gain?

2. The RIN parameters are not given, 

3. we do not know any about the phenotypic trait after these two feeding systems. 

4. no information about sequencing - pairend single and, how many cycles?

5. in the abstract was pinpointed thatDEGs by log2FC>0  and in the methods log2FC>1. ??

6. In the introduction are repeated sentences.

7.  the study lacks of validation stage by using for example qPCR method which confirms the obtained RNA-seq results.

8.  In the discussion and conclusion are given information that it was analyzed liver transcriptomes before and after fatting, which is no wrigth because it was compared the two transcriptomes with two feeding systems.

 Because of so many mistakes and errors and no validation control, I propose to reject this manuscript. The English language also needs thorough corrections.